# Multi-model methods for structural analysis of China's green economy network based on input-output method

**Yigang Guo[1], Shaoling Ding[1,2]\*, Jingliang Huai[1], Jiayao Pan[1], Yan Meng[3]**

**1** School of Mathematics and Statistics, Guilin University of Technology, Guilin, China, **2** Guangxi Colleges and Universities Key Laboratory of Applied Statistics, Guilin, China, **3** College of Foreign Studies, Guangxi Normal University, Guilin, China

\* dingshl@glut.edu.cn

**Data Availability Statement:** The data underlying the results presented in the study are available from China Bureau of Statistics.All original data in the paper are downloaded from Zenodo database (https://zenodo.org/records/10410182).

## Abstract

The green economy has been advocated globally as a solution to environmental issues. In China, it is considered a national strategy for future economic development. This study utilizes methods such as Industry Network, Maximum Spanning Tree (MST) method, Leiden Community Clustering (LCC) algorithm, and Weaver-Thomas (WT) model to explore the contribution and position of the green economy and industries in China's economic development. The findings are as follows: (1) The density of China's green industry network has experienced a process of initially tightening and then loosening, ultimately tending towards stability. (2) The trunk structure of China's industrial network remains relatively stable, forming an industrial structure with electricity, heat production and supply as the core. (3) China's industrial and green industry communities continue to improve and become more cohesive, but some green industries are still on the periphery of communities. (4) The ability of green industries to pull other industries is weak, and the subsequent promotion momentum needs to be improved. However, the green industry still has enormous room for growth and potential to unleash its long-term positive multiplier effects. More attention and support need to be given by managers and decision-makers, so that it can make better contributions to society and the economy.

## 1.Introduction

Since the 21st century, global environmental issues such as depleting resources, frequent extreme weather events, threats to biodiversity, and a deteriorating ecological environment have become increasingly prominent. In response to these challenges, the concept of a green economy has gained widespread attention as a new economic model aimed at environmental protection. With China's latest industrial upgrading and economic transformation, many industries, led by traditional industries such as agriculture, automobile manufacturing, transportation, and energy, have embarked on a green shift, and many new economic forms have emerged. As shown in Fig 1A, China's Green Development Index has improved from 53.01 in 2016 to 70.12 in 2020, an increase of 32.23%. In addition, the secondary indicators of the Green Development Index all have significant improvements of different degrees. The

**Funding:** This research is funded by the National Natural science Foundation of China, grant number 72263004; and the Guangxi Natural Science Foundation, grant number 2023GXNSFBA026171. The funders had no role in study design, data colection and analysis, decision to publish, opreparation of the manuscript.

**Fig 1. China's green economy indicators index.**

development of green economy proposes solutions to China's environmental problems and becomes an important driving force for China's economic development mode and industrial transformation and upgrading [1]. The development of the green economy provides solutions to China's environmental problems and has become an essential driving force for China's economic growth and industrial transformation.

Since the 18th Party Congress of the CPC, the Chinese leadership attaches great importance to the comprehensive green transformation of economic and social development. It requires all regions to adhere to the development path of ecological priority and green low-carbon, strives to promote a green and low-carbon economy and realize the modernization of the harmonious coexistence of human beings and nature. Mr.Xi has made essential instructions at important occasions such as the United Nations General Assembly, the World Economic Forum, and the CPC's 20th Party Congress to adhere to green development, deeply grasp the green low-carbon and recycling development as a new economic growth point, and pointed out that it is necessary to accelerate the promotion of scientific and technological revolutions in the field of the green economy, to promote the comprehensive green transformation of China's industries, and strive to realize China's "dual-carbon" goal. In addition, the CPC and the State Council have introduced a series of policies and measures to enhance people's awareness of conservation, environmental protection, and ecology, including garbage classification, low-carbon travel, and the promotion of clean energy. In February 2021, the State Council issued the "Guiding Opinions on Accelerating the Establishment of a Sound and Efficient Green, Low-Carbon, and Circular Development Economic System," which clarified the relationship between various industries and green economy and further established a policy system to support the development of green economy from an institutional perspective. These measures indicate that China is gradually balancing and coordinating the relationship between economic growth and ecological and environmental protection, supporting the development of green industry and green economy with scientific and technological progress and policy system, and promoting high-quality economic development.

Research has rapidly expanded as the green economy becomes a new engine for China's economic development. Some scholars believe that the green economy, as a new financial

form, is a powerful engine and support for achieving green growth and a new way to transform China's development model and solve the challenges of economic development and environmental protection [2,3]. There are also scholars that China's green economy development should take into account its advantages and disadvantages to contribute to China's sustainable development [4]. Other scholars have analyzed the existing demands of China's green economy and the insights from international organizations such as the United Nations, proposing theoretical models and future development paths for China's green economy [5]. Regarding the research status of China's green economy, scholars have adopted different methodologies. Some have utilized panel data from various provinces in China to establish a low-carbon indicator system for evaluating green economic efficiency at the regional level [6]. Others have conducted comprehensive analyses of the green economy through econometric approaches, examining the impacts of economic development level, technological innovation, industrial structure, and foreign direct investment (FDI) on the green economy [7,8]. Moreover, they have investigated the relationship between the green economy and local policies [9]. At the international research level, scholars have examined the evaluation of the green economy in rural areas of Poland, taking into account environmental, economic, social, and agricultural factors [10]. Furthermore, some scholars study the green economy policies of some developing countries and the urgency and necessity of developing a green economy in the world and Russia [11,12].

In this paper, by combing and summarizing the existing research on the green economy, scholars' research on the green economy mainly focuses on two aspects. On the one hand, they explore the theory and practice of the green economy to provide insights and guidance for its promotion and implementation. On the other hand, scholars have been committed to constructing evaluation indicator systems. They either measure the regional variability of the green economy from different indicators or estimate the scale of the green economy from an accounting viewpoint. These studies have made significant contributions to a deep understanding of the development of China's green economy. Given the current importance of the green economy as a development strategy in China, this motivates us to analyze the position of the green industry in the structure of industry China's industry, and to accelerate the construction of modernization in which human beings live in harmony with nature.

Currently, it is the Input-Output Table (IOT), created by Leontief, that can systematically and sustainability reflect internal linkages among industries within a region [13,14]. The first quadrant of IOT records the input and output of intermediate products in the production processes of different industries. The horizontal direction reflects the allocation of products in various sectors, and the vertical direction reflects each sector's consumption of other products. The IOT can document the linkages between industries, especially the indirect effects of demand changes in one sector on others. Additionally, IOT adheres to certain balance relationships and has become a primary means for scholars to study industrial structures. Meanwhile, it has developed many research methods based on IOT, such as Input-Output Analysis (IOA), Data Envelopment Analysis (DEA), and Structural Decomposition Analysis (SDA) [15–18].Scholars have utilized IOA to analyze the structure of China's energy industry [19,20]. Some scholars have used the DEA model and output index of relative green GDP to assess the Green Economic Efficiency (GEE) of 29 provinces in mainland China from 1995 to 2007 [21]. Some scholars have measured China's provincial GEE from 1995 to 2012 based on the improved DEA-SBM model to explore practical ways to improve GEE [22]. Additionally, there have also been studies that use pre-2009 U.S. input-output data to allocate carbon emissions from different types of energy consumption and use SDA to decompose the socioeconomic factors influencing carbon emissions changes [23]. As the development of industrial production and inputs takes on a networked character, the supply of the main factors of

production (whose payment is income in several forms, such as salaries, profits, and rents) is made possible through trade activities to downstream resources along the supply chain [24]. Therefore, scholars combine input-output analysis and complex networks to analyze the industrial structure.

A complex network is a quantitative analysis method based on the relationships between nodes and edges. It integrates mathematical methods and graph theory to reveal relationship characteristics among individuals in the network. From the analysis perspective, complex networks can be utilized to calculate the association characteristics of the entire network, study the distribution of network structure, and analyze the network status of a node. Due to their utility and interpretability, network analysis has been widely applied in multiple research domains, such as epidemic spread, public transportation, development evaluation systems, clinical medicine, etc [25–28].The input-output system is a typical network system that treats each industry as a "node" and economic exchanges between industries as a "edge." The network analysis can be applied to IOT to explore internal industrial linkage characteristics comprehensively. From a global perspective, scholars have analyzed the global energy structure based on IOT and revealed the elements shown by hidden energy flow networks. Scholars have also studied the global carbon footprint and conducted network visualization and econometric analysis. From a regional research perspective, scholars have used network analysis based on Chinese input-output data to study the energy-water-CO2 metabolism system in China, identify the industries with the highest energy consumption, and provide recommendations for energy conservation and emission reduction [29]. In addition, researchers have conducted studies on the carbon emission system in India based on the Input-Output (IO) and Social Accounting Matrix (SAM) frameworks [30]. Then, they have also employed Structural Path Analysis (SPA) to analyze the differences in transmission mechanisms between these two frameworks. By the network analysis method, the research perspective of the input-output system is extended, allowing for the comprehensive excavation of the characteristics of input-output networks from a holistic to a local level. It is imperative to note that the input-output network is a complex network with both directions and weights. However, for different research purposes, some studies have simplified the network structure of input-output, such as transforming it into an undirected network or an unweighted network that does not take into account the data information take into account inter-industry data information.

In the network, what is the position of the green economy and its related industries in China's economic development? What direct and indirect effects exist among these industries? What are the industrial community effects in the green economy industry? Under the current financial foundation, which specific areas of development within the green economy are more conducive to economic growth? To solve these problems, this paper transforms the complex influence effect relationship between industries based on the input and output data in different periods into the data structure of a complex network. It constructs a dynamic industrial network model to analyze the characteristics and development trends of the industrial linkage structure demonstrated by the green economy. Due to the multidimensional nature of the network structure, we analyze the industry structure through overall metrics and node metrics. The overall characteristics are measured by topological properties such as network density and average shortest path. The MST method is used to find the backbone network in the industry and analyze the correlation shown by each industry node in the network and the community structure based on the network graph. In addition, LCC algorithm is introduced as a complementary indicator of the nodes to analyze the role played by the green economy and its industries in the community. Considering the heterogeneity of nodes, the importance of each indicator is uncertain. Therefore, we use the WT model on the basis of entropy weight method

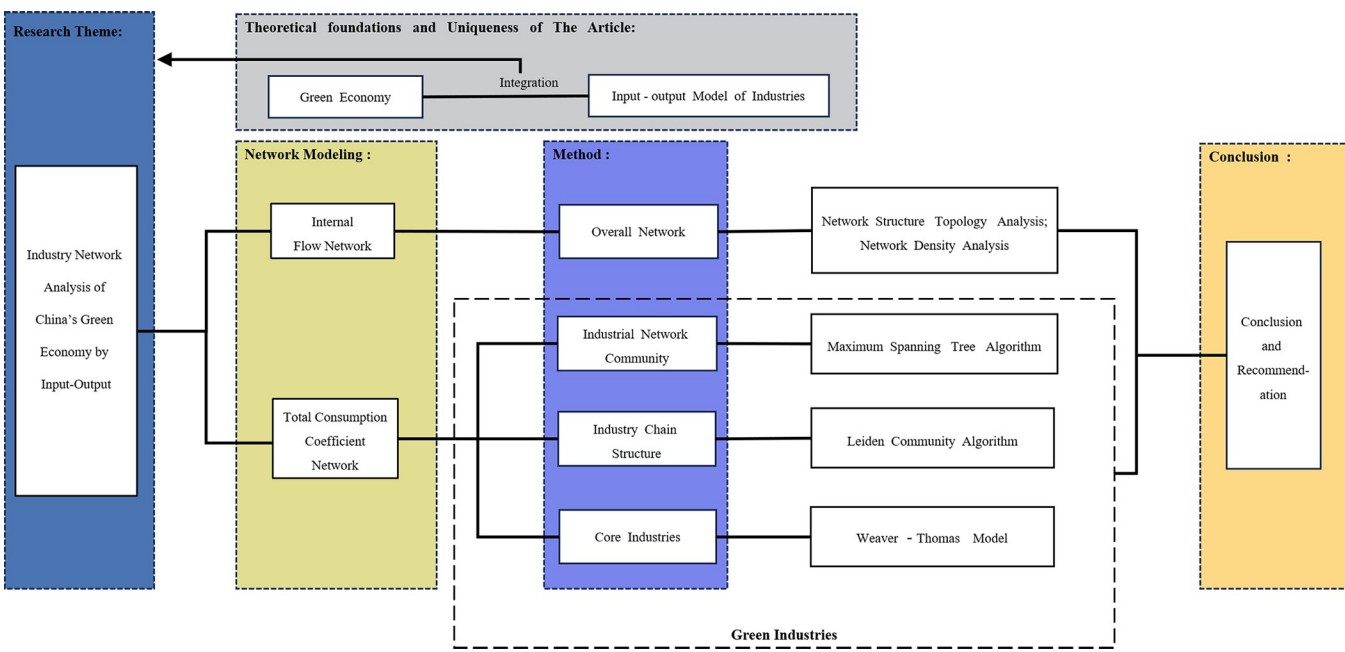

**Fig 2. Research roadmap.**

to get the importance ranking of industrial nodes and analyze the position and role of green economy in China's national economic system.

Compared with previous studies, the main contributions of this paper are as follows. Firstly, it extends the literature on industrial network building. Compared with the previous literature, this paper lays a good foundation for future research in building network models from the perspective of complex networks on the data of IOT and identifying the industrial network backbone through the MST method. Secondly, it provides a comprehensive evaluation method for green industry analysis by combining the entropy weight method and WT model. The study characterizes the green economy and its industries at the whole and part levels, respectively. Thirdly, the importance ranking of industry nodes provides a theoretical basis for policy formulation and industrial development direction of green economy. The A1- Agricultural products and C4- Electricity, heat production and supply occupy an important place in the industrial structure. The Chinese government should invest in and support the development of the Ecology and Energy saving and environmental protection industries. The research roadmap of this study is shown in Fig 2.

The structure of this paper is organized as follows: The Definition of Green Economy section explains the green economy from an industrial perspective. Then in the Green economy industry network analysis methods and data sources section presents the methods and data used to analyze the green economy industry network. An analysis of the green economy industry network is conducted in the Green economy industrial network analysis system section. Finally, in the Conclusion and Recommendations section, we present the conclusion and recommendations.

## 2.Definition of green economy

### 2.1 The meaning of the green economy

The concept of the green economy was first introduced in 1989 by British economist David Pearce in "Blueprint for A Green Economy." The green economy is the economic form of

green development, mainly composed of energy-saving and environmental protection industries and low-carbon industries, also known as ecological protection or low-carbon economies. Since then, numerous scholars and organizations have put forward their views on the focus of the green economy concept. In the 1990s, the focus of the green economy concept was directed toward the economic instruments of pollution control by scholars such as Michael Jacobs. The wealth value of the ecosystem is emphasized, with the goal of ecosystem protection remaining unchanged, which is essentially a further extension of traditional environmental protection methods. Twenty years after the concept of sustainable development was put forward, people's thinking about the green economy is more comprehensive than just pure ecological governance. Instead, it has started to surpass the conventional boundaries of environmental economics, focusing on the overall greening and decarbonization of the economy while combining economic growth with ecological conservation. With the financial and climatic changes, the green economy has been endowed with more significant connotations by people. The UNEP definition of a green economy is "well-being and social equity while significantly reducing environmental risks and ecological scarcities" [31]. At this time, attention to the green economy is more focused on promoting social well-being, emphasizing a model of sustainable development with social equity based on minimizing environmental risks and ecological damage [32,33].This paper focuses more on recent research related to China's green economy in recent years. Based on the theory of "green mountains are golden mountains" in China, the concept of the green economy is derived, that is, "more emphasis on ecological priority, and at the same time, require that the linkages between economic, social and natural systems are more systematic, holistic and coordinated."

## 2.2 Definition of green economy industries

Scholars at home and abroad define green industry from the following perspectives: sustainable development, industrial composition, emission reduction, and the combination of different benefits and output [34,35]. According to the definition given by the International Green Industry Union (IGIU), green industry refers to industries that, in their production processes, prioritize environmental considerations and utilize technology to implement green production mechanisms aimed at conserving resources and reducing pollution (energy conservation and emissions reduction). However, due to the significant differences in the industrial structure of each country, there has yet to be a unified international standard for the classification of green industry. Drawing on the "National Economic Industry Classification" (GB/T4754-2017) issued by the National Bureau of Statistics of China and the concept of the green economy, this article incorporates the availability of comprehensive data and the recognizability of specific characteristics. And regarding the "Green Industry Guidance Catalog (2019 Edition)", the green industry was finally specifically subdivided into four representative business types: ecology, energy saving and environmental protection, clean energy and pollution control, green infrastructure upgrades and services. The ecology includes agriculture, forestry, animal husbandry and fishery professional and auxiliary activities, agriculture, forestry, animal husbandry, and fishery. These industries aim to provide raw materials such as food, timber, and other natural resources, continuously improve the ecological industry's benefits, and offer new means and tools for environmental protection and sustainable development. Energy saving and environmental protection correspond to five categories in the "National Economic Industry Classification" (GB/T4754-2017): general equipment manufacturing, electrical machinery and equipment manufacturing, automobile manufacturing, instrument and apparatus manufacturing, and comprehensive utilization of waste resources. These industries provide facilities and technologies for

**Table 1. The classification of green economy business types and their corresponding industry classifications.**

| Green Industry Guidance Catalog | National Economic Industry Classification 2017 | Input-output table industry classification 2020 | Code name |
|---|---|---|---|
| Ecology | 01 Agriculture | 01 Agricultural products | A1 |
| | 02 Forestry | 02 Forest products | A2 |
| | 03 Animal husbandry | 03 Livestock products | A3 |
| | 04 Fishery | 04 Fishery products | A4 |
| | 05 Agriculture, forestry, animal husbandry and fishery professional and auxiliary activities. | 05 Agriculture, forestry, animal husbandry and fishery service products | A5 |
| Energy saving and environmental protection | 34 General equipment manufacturing | 65 Boilers and prime mover equipment | B1 |
| | 38 Electrical machinery and equipment manufacturing | 76 Auto Parts and Accessories | B2 |
| | 36 Automobile manufacturing | 78 Ships and related installations | B3 |
| | 40 Instrument and apparatus manufacturing | 80 Motor | B4 |
| | 42 Comprehensive utilization of waste resources | 81 Power transmission and distribution and control equipment | B5 |
| | | 83 Battery | B6 |
| | | 92 Instrumentation | B7 |
| | | 94 Waste resources and waste materials recycled into processed products | B8 |
| Clean energy and pollution control | 35 Equipment manufacturing | 74 Other specialized equipment | C1 |
| | 76 Water management | 125 Water management | C2 |
| | 44 Electricity and heat production and supply | 126 Ecological protection and environmental governance | C3 |
| | 77 Ecological protection and environmental control | 96 Electricity, heat production and supply | C4 |
| Green infrastructure upgrades and services | 37 Railway, ship, aerospace, and other transport equipment manufacturing | 77 Railway transportation and urban rail transit equipment | D1 |
| | 50 Building decoration, renovation and other construction industries | 102 Building decoration, renovation and other construction services | D2 |
| | 78 Public facilities management | 127 Public facilities and land management | D3 |
| | | 113 Catering | D4 |
| | | 128 Resident Services | D5 |

each field's green and sustainable development, reducing resource consumption and environmental pollution. Clean energy and pollution control is comprised of ecological protection and environmental control, special equipment manufacturing, electricity and heat production and supply, and water management, replacing the original high energy consumption resources and providing environmentally friendly clean energy. Green infrastructure upgrades and services mainly represent the industries for establishing urban space for the harmonious coexistence of man and nature and improving people's "sense of green access," corresponding to the three significant sectors of railway, ship, aerospace, and other transport equipment manufacturing, building decoration, renovation and other construction industries, and public facilities management. The industry classification of the input-output table data released by China is similar to the National Economic Industry Classification, but the category is more detailed. These classification methods can be applied to the four mentioned types of green industry business, as shown in Table 1. The classification of green economy business types and their corresponding industry Table 1. The imputed data of historical years can be better calculated and analyzed by de-fining and dividing the layout of business types. It also reflects the unique categorization nature of China's green economy and enables adequate delineation and subdivision of similar economic concepts.

## 3.Green economy industry network analysis methods and data sources

The industrial network analysis method is based on social network analysis, input-output analysis, and the data structure and analysis process of complex networks, which analyses the association, structure, and resource mobility among the participants in the network in the form of a network [36,37]. As a type of industry associated with various types of industries, the green sector requires the utilization of industrial network approaches to analyze its mode of operation, industrial driving force, and importance.

### 3.1 Construction of input-output-based industrial network

Input-output analysis is a quantitative economic analysis method that reveals the complex interdependencies and interrelationships between industrial sectors connected by sectoral monetary transactions. It can be used to study the interdependencies between other sectors within an economic system, focusing on the relationship between inputs and outputs [38]. The core of the method is the IOT created by Leontief, an American economist and Nobel Prize winner in economics. This article constructs an industrial network for analysis based on the original flow matrix and the total consumption matrix of the IOT.

**3.1.1 Internal flow network matrix.** The internal flow matrix in the IOT runs through the production process of the whole economic sector, reflecting the connection between industrial sectors in the production process. The complex industrial network primarily originated from collecting industries, a method for studying the links between sectors within a system. The strategy is based on the internal flow matrix $M_{n\times n}$ and sets the network structure as $G = (V,E,W)$, where $V = \{v_1,\cdots,v_n\}$ denotes the individual industries in the flow matrix ("nodes"), $E = \{e_{ij}\}(i,j = 1,\cdots,n)$ represents the inter-industry flows ("edges"), and $W = \{w_{ij}\}(i,j = 1,\cdots,n)$ is the magnitude of the inter-sectoral linkage flows ("weights "). It is important to note that $w_{ij}$ indicates the amount of value consumed by industry sector $v_j$ to industry sector $v_i$ if the linkage flow $w_{ij}$ is 0, $e_{ij} = 0$; at this point, there is no linkage between sectors. Otherwise, $e_{ij} = 1$ indicates that there is a linkage between sectors.

**3.1.2 Complete exhaustion coefficient network matrix.** In academia, the input-output analysis method is typically employed based on the IOT to describe the resource input and output relationships between different industrial sectors within an economy. The method assesses the linkage between different industries through an input-output model. Considering the complex relationships between industries, these associations can be categorized as direct or indirect connections, represented by direct and indirect consumption coefficients. Thus, the total consumption coefficient is selected to measure the inter-industry connection. The total consumption coefficient is obtained by adding the direct and indirect ones. It comprehensively analyzes the relationships between different industries, considering the entire economy's complexity. Therefore, it accurately captures the extent of interdependence among industries regarding their utilization of intermediate products. In summary, the interaction relationship between each industry sector is expressed as a matrix by constructing the total consumption coefficient matrix. The total consumption coefficient matrix $B$ is calculated as follows:

$$\begin{cases} a_{ij} = \dfrac{x_{ij}}{X_j}(i,j = 1,2,\cdots,\text{n}) \\[2mm] A = \begin{pmatrix} a_{11} & \cdots & a_{1n} \\ \vdots & \ddots & \vdots \\ a_{n1} & \cdots & a_{nn} \end{pmatrix} \\[2mm] B = (I-A)^{-1} - I \end{cases} \qquad (1)$$

Among them, $a_{ij}$ represents the direct consumption coefficient, $A$ represents the direct consumption coefficient matrix, $I$ represents the unit matrix, $(I–A)^{-1}$ represents the Leontief inverse matrix $L$, and $B$ is the total consumption coefficient matrix. Similarly, by referring to the method of the internal flow matrix, the network structure of the total consumption coefficient matrix, known as $G_B = (V_B, E_B, W_B)$, is constructed.

## 3.2 Characteristic index analysis system

Based on the above two industry networks, this paper applies three methods to analyze the green economy industry: the MST method, the LCC algorithm, and the WT model. Utilizing these three methods helps provide a more comprehensive analysis of the interaction between the green economy industry and other industries and identifies the dominant industries within the network. The main algorithms and network analysis indicators used above are as follows:

**3.2.1 The most vital industry chain path–MaxTreeA.** The network structure can be messy, which may obscure meaningful connections between nodes in the network diagram, for example, with many edges. Scholars have traditionally utilized the Minimum Spanning Tree algorithm to generate a new industrial network, which enables uncovering the original network's topological properties and clustering structure. According to the idea of a minimum spanning tree and the Kruskal algorithm modification is made by retaining the original smaller weight between nodes to retaining the more significant weight between nodes [39,40]. At the same time, the directed right in the network is changed to undirected to reflect the most vital industrial chain path in the network, that is, to form the MST method that can extract the network's backbone. In this paper, we draw on and readjust the methodology to reshape the network subgraph *TreeA* : $G_{Tree} = (V_{tree}, E_{tree}, W_{tree})$ under the fully consumed coefficient network matrix $G_B = (V_B, E_B, W_B)$.

The specific operation to transform $G_B$ into $G_{Tree}$ is as follows:

Compare the fully expended coefficient matrix $b_{ij}$ with $b_{ji}$, select the larger values, and overwrite the smaller values, forming a new matrix $B'_{n \times n}$;

Weights of all directed edges of in descending order;

Select the edge with the maximum weight and its two endpoint nodes as input for matrix $G_{TreeA}$;

Iteratively add edges until the graph represented by matrix $G_{TreeA}$ is fully connected;

Check whether the number of edges of the graph $G_{TreeA}$ is less than $N–1$; if the number of edges is less than $N–1$, then continue to repeat the above steps.

If the selected edge causes graph *TreeA* to have a self-loop on a node, discard that edge and select a new one. Eventually, form a connected graph *TreeA*, denoted as $G_{TreeA}$.

**3.2.2 Industrial network community analysis—leiden community clustering algorithm.** An industrial community can be regarded as a network consisting of many nodes. In the industrial community network, there is a difference in the sparseness of the links between different industrial nodes. The density of these connections impacts the collaboration and interconnection between industries and manifests in the network structure, giving rise to various network communities. By applying relevant algorithms, industries with tightly-knit internal relationships can be categorized into the same industry community. Community Detection is commonly used as an algorithm to analyze large and complex network structures by clustering the associated nodes of a large amount of data to find the nodes with similar properties and close ties and divide them into non-overlapping communities. This process helps depict the internal structure and composition of the network, providing an objective way to reveal its internal organization, connections, and characteristics. This paper adopts the LCC algorithm, which is optimized based on the Louvain algorithm and integrates the ideas of Smart Local Move, Fast Local Move, and Random Neighbor Move.

In addition, the algorithm overcomes Louvain's shortcomings and ensures connectivity between nodes in the community while obtaining more stable and high-quality partitions [41,42].The performance of LCC algorithm is usually measured by using the Modularity metric, which is calculated by the following formula:

$$
\begin{cases}
Q = \dfrac{l}{2m} \sum_{ij} (w_{ij} - \dfrac{w_i w_j}{2m}) \delta(c_i, c_j) \\
\delta(c_i, c_j) = \begin{cases} 1, & i = j \\ 0, & \text{else} \end{cases}
\end{cases}
\tag{2}
$$

Above, $Q$ denotes the modularity degree, $i$ and $j$ represent any two nodes in the network; $w_{ij}$ is the weight of the connecting edges between node $i$ and node $j$, $w_i = \sum w_{ij}$ is the sum of the weights of all the connecting edges of node $i$, and $w_j$ s the sum of the weights of all the connecting edges of node $j$; $c_i$ and $c_j$ denote the communities that contain nodes $i$ and $j$, respectively. $\delta$ $(c_i, c_j)$ is a function that evaluates whether nodes $i$ and $j$ belong to the same community. If they do, the output is $\delta(c_i, c_j) = 1$; otherwise, it is $\delta(c_i, c_j) = 0$.

**3.2.3 Strong industry extraction—Weaver-Thomas model based on entropy weighting approach.** eeThe MST method is used to clarify the network, while the WT model is used to select the dominant industry. The basic principle is to compare an observed distribution with an assumed distribution to construct the closest approximation. When applying this model to industrial selection, the first step is to sort the indicators in ascending or descending order. Then, calculate and compare the sum of squared differences between each hypothetical distribution and the actual distribution to determine the best fit. A smaller sum of squares indicates that the hypothetical distribution closely approximates the optimal observed distribution [43,44]. The threshold is an approximation of the optimal distribution and shows how many values to choose to get as close as possible to the hypothetical distribution. One of the most critical aspects for WT model is the selection of the weights of each evaluation metric. In order to reduce the influence of subjectivity on the decision-making results and present the original data's information more objectively, the principle of entropy is adopted in this paper to explain the amount of information of each index. Entropy can measure the degree of disorder in a system, and entropy weight reflects the amount of adequate information carried and transmitted by each indicator of the system. In other words, the more helpful information an indicator carries and transmits, the higher its entropy weight, and vice versa [45–48]. The weights of the indicators are finally derived after correction, which improves the rationality and precision of the indicator assignment. The detailed construction steps are as follows:

Firstly, it is stipulated that the matrix of total consumption coefficients in the IOT is:

$$
B = (b_{ij})_{n \times n}
\tag{3}
$$

Among them, $b_{ij}$ represents the value of the $j$–th indicator for the $i$–th industry, and $n$ represents the dimension of the industry network matrix corresponding to the IOT. The matrix of total consumption coefficients can reflect the production technology and economic links between industries, which helps us to understand the consumption status of products between industries more clearly and thoroughly and analyze the correlation characteristics between industries. The specific calculation process is as follows:

Step 1. Calculate the weights of each evaluation index. First, calculate the weight of indicator $P_{ij}$.

$$P_{ij} = \frac{b_{ij}}{\sum\limits_{i=1}^{n} b_{ij}} \tag{4}$$

Then, the entropy value $e_j$ and the coefficient of variation $g_j$ were calculated for each indicator.

$$\begin{cases} e_j = -\dfrac{1}{\ln(n)} \sum\limits_{i=1}^{n} P_{ij} * \ln P_{ij} \\ g_j = 1 - e_j \end{cases} \tag{5}$$

Finally, calculate the weights of specific $j$–th indicators $d_j$.

$$d_j = \frac{1 - e_j}{\sum\limits_{j=1}^{n}(1 - e_j)} = \frac{g_j}{\sum\limits_{j=1}^{n} g_j} \tag{6}$$

Step 2. Build the WT model. Firstly, based on the full depletion matrix, the WT index is used to screen the industries to obtain effective correlation thresholds. The expression for the WT index ($WT_{ij}$) after sorting the values of indicator $b_{ij}$ in ascending order is:

$$\begin{cases} WT_{hj} = \sum\limits_{i=1}^{m}\left(U_i^h - \dfrac{100 b_{ij}}{\sum\limits_{i=1}^{m} b_{ij}}\right)^2 \\ U_i^h = \begin{cases} \dfrac{100}{h}, i \le h \\ 0, i > h \end{cases}, h = 1, 2, \cdots, n \end{cases} \tag{7}$$

Secondly, $WT_{hj}$ is calculated for all industry sectors under indicator $j$–th, and these values are minimally filtered to obtain $minWT_{hj}$. When $WT_{kj} = minWT_{hj}$, the position count $k$ at which the minimum the $j$–th indicator $WT_{hj}$ combination occurs, is determined at that time, representing the number of dominant industries under each indicator. Subsequently, taking the average of these counts, we obtain the total number of dominant industries, $q$.

$$\begin{cases} q_j = \{k; WT_{kj} = minWT_{hj}, k = 1, 2, \cdots, n\} \\ q = \sum\limits_{j=1}^{n} \dfrac{q_j}{n} \end{cases} \tag{8}$$

Then, a composite ranking matrix $R$ is constructed and based on the weights of the

indicators, and the $i$−th industry composite ranking value $R_i$ is calculated.

$$
\begin{cases}
R = \begin{pmatrix} R_{11} & \cdots & R_{1n} \\ \vdots & \ddots & \vdots \\ R_{n1} & \cdots & R_{nn} \end{pmatrix} = \{R_{ij}\}_{n \times n} \\
R_i = \sum_{j=1}^{n} d_j R_{ij}
\end{cases}
\tag{9}
$$

In the formula, $R_{ij}$ represents the ranked value of the $j$−th indicator corresponding to the $i$−th industry, which can be positive or negative, and $d_j$ represents the weight value of $j$−th indicator. After the comprehensive ranking value $R_i$ is sorted from high to low, the top industries (not more than $q$) are selected as the leading industries.

### 3.3 Data sources and data processing

In this paper, the input-output data released by the National Bureau of Statistics of China is selected as the basis for constructing the industrial network to reflect China's industrial and economic structure comprehensively. China's IOT is a set of authoritative data released by the National Bureau of Statistics of China, which is used to describe the input-output relationship among Chinese industries. The internal flow matrix reflects the supply and demand relationships between different industries and can accurately show their intricate connections [20,49].

China's official IOTs for 2012, 2017, 2018, and 2020 are surveys with a more detailed industrial categorization. For other years like 2010 and 2015, the IOTs are estimated based on the previous survey results, and the industry classification is limited to 42 sectors. Hence, in this study, the input-output tables of China for 2012, 2017, 2018, and 2020 are taken as the foundational data. The industry quantities in each IOT are standardized into the same 139 industry categories based on the overall characteristics of the network and the analysis results of the industry network, concerning the "National Economic Industry Classification" (GB/T4754-2017). Subsequently, an industry network is constructed for computational analysis.

## 4.Green economy industrial network analysis system

### 4.1 Industrial networks based on inputs and outputs

Using the internal flow array $M$ of China's input-output analysis, a 139×139 network matrix of four annual industrial flow relationships was formed, and the complex industrial network $G$ was shaped separately. The results obtained are visualized in Fig 3.

The relationship of the industry network is shown in Fig 3 by Gephi 0.9.7, in which the relationships between nodes in the industrial complex network are tightly connected, but the visualization effect is not significant. Further quantification and structural analysis are needed. The nodes in the four networks of the appeal are all 139, including five nodes of the ecology, eight nodes of the energy saving and environmental protection, four nodes of the clean energy and pollution control, and five nodes of the green infrastructure upgrades and services.

Based on Table 2, by examining the topological properties of the network, it is found that the average path lengths of the 2012 network and the three subsequent networks after 2017 are 1.254 and 1.243, respectively. The decrease in the distance between any two industries indicates a tighter linkage between nodes in the industry network and faster and more efficient information dissemination. This is because after 2016, China guided the development of low-carbon industries, actively explored the transformation of traditional industries, and promoted

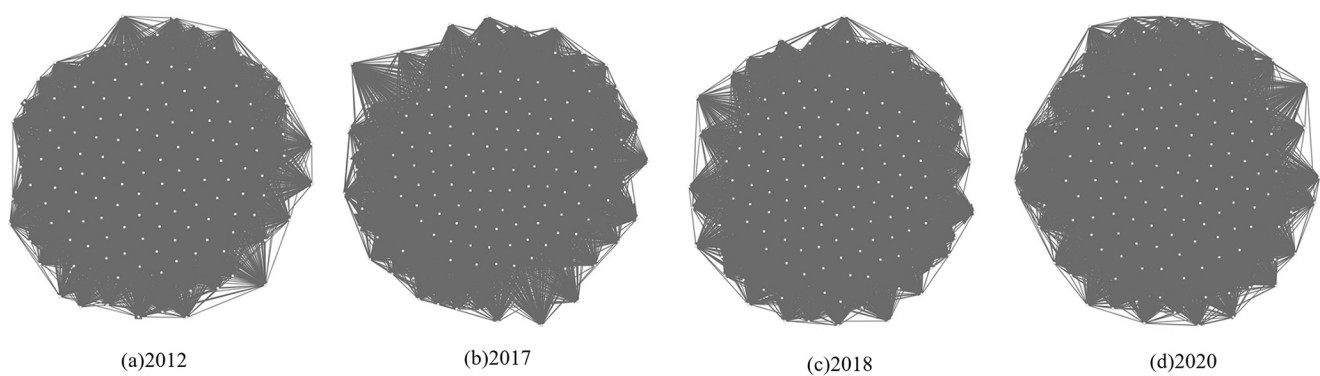

(a)2012 (b)2017 (c)2018 (d)2020

**Fig 3. Industry network timing diagram.**

the modernization of industrial chains with new technologies [50]. In addition, according to data from China's Ministry of Commerce, the proportion of foreign investment in high-tech industries has increased from 14.1% in 2012 to 36.1% in 2022, an increase of more than one times. During this period, the structure of China's actual use of foreign investment was gradually upgraded, and the amount and level of foreign direct investment increased significantly. This move can promote economic cooperation, technological innovation, and resource sharing between regions and countries. It can help China realize the modernization and upgrading of the industrial chain as soon as possible [51]. Additionally, the clustering coefficients of the networks are 0.955 and 0.944, respectively, which further demonstrates that the links in the network are closely connected. However, the clustering coefficient in 2017 and after decreased compared to 2012, mainly due to the outbreak of the subprime mortgage crisis in the United States in 2008. The outbreak of the financial crisis hit the global economic market, and the global economy suffered a severe downturn, and China was inevitably affected [52]. However, in general, its network, has the characteristics of a high clustering coefficient and shorter average node characteristic path length, which is in line with the characteristics of the small-world network [53].

From the perspective of the change of the number of network connection edges in the IOT in Table 3, the number of input-output network connection edges decreased from 14323 to 13896 from 2012 to 2017. It indicates decreased inter-industry connections, suggesting stricter communication between related industries and a greater emphasis on intensive and vertical production linkages. The production process becomes more refined and specialized. The characteristics of the industrial network showed smoothness after 2017, indicating that the linkages between industries tend to be constant, and the division of labor among industries is more straightforward and standardized. Similarly, the green industry network connecting edges show the same change trend as the number of connecting edges of the overall network. The network connecting edges of the ecology type decreased from 366 in 2012 to 245 in 2017, and it has remained consistent with the number of edges in the ecological and environmental industry network for 2018 and beyond. Moreover, the remaining three types of green industry networks exhibit similar changes, with the number of edges initially decreasing and stabilizing.

**Table 2. Topological nature of the network.**

| Character / Year | 2012 | 2017 | 2018 | 2020 |
|---|---|---|---|---|
| Average path length | 1.254 | 1.243 | 1.243 | 1.243 |
| Clustering coefficient | 0.955 | 0.944 | 0.944 | 0.944 |

**Table 3. The number of edges connected between the whole industry network and the green type industrial network.**

| Year<br>Number | 2012 | 2017 | 2018 | 2020 |
|---|---|---|---|---|
| Holistic network | 14323 | 13896 | 13898 | 13898 |
| Ecology | 366 | 245 | 245 | 245 |
| Energy saving and environmental protection | 959 | 832 | 833 | 833 |
| Clean energy and pollution control | 532 | 428 | 428 | 428 |
| Green infrastructure upgrades and services | 640 | 572 | 572 | 572 |

Overall, the number of connection edges in the Green Economy Industry Network is trending down faster than the average for the network as a whole. Among them, the number of connected edges of the ecology network declined the fastest, reflecting a more professional and standardized level of connectivity in this network [1].

## 4.2 Strongest backbone network

Based on the IOT, the number of nodes in the industrial complex network reaches 139. If any two nodes have connecting edges, the maximum number of connecting edges that may appear in the network is 19,182. According to Fig 3, the number of connecting edges obtained is also more than 13,000, which shows that the relationship between the nodes of the industrial complex network is very close. In general, among the many industrial linkages, there are necessarily strong and weak connections. In general, among the many industrial linkages, there are necessarily strong and weak connections [54].Given the network's denseness, the industry's strong connections are screened to more clearly show the characteristics of the industrial linkage structure shown in the IOT.

**4.2.1 Industrial backbone network extraction.** Based on the total consumption coefficient matrix of the IOT in 2012, 2017, 2018, and 2020, the industrial network is constructed, and the MST method is used to extract the structure of the backbone network. The main idea of the MST method that by retaining the connecting edges and their nodes with larger weights in the network and eliminating the connecting edges with smaller weights and weaker connecting relationships. It ultimately results in the computation of an undirected tree graph without cycles, as shown in Figs 4–7.

As can be seen from Figs 4–7, the backbone structure of China's industrial network in 2012–2020 remained relatively stable, but some changes still occur within localities. Specifically, the most central industry in the network is C4-electricity and heat production and supply, with the most significant number of associated nodes. The following prominent industries include A1-agricultural products, basic chemical raw materials, and non-ferrous metals and their alloys. China's central industries are mainly distributed in the secondary industry, because China is the world's largest manufacturing country, and the industrial output value occupies an important position in the national economy. These industries form a high degree of resource dependence on other industries [55]. These central industrial nodes have more links with other industries in the industrial network, and an industrial community with it surrounds each central node as its core. These gathered industrial communities either have homogeneity with each other or show upstream and downstream relationships in the industrial chain, indicating a robust industrial support network.

**4.2.2 Industrial network community analysis.** Different solid and weak associations between industries form different group relationships, manifesting as industrial communities in economic activities. The communities existing in the network with aggregation effect can be

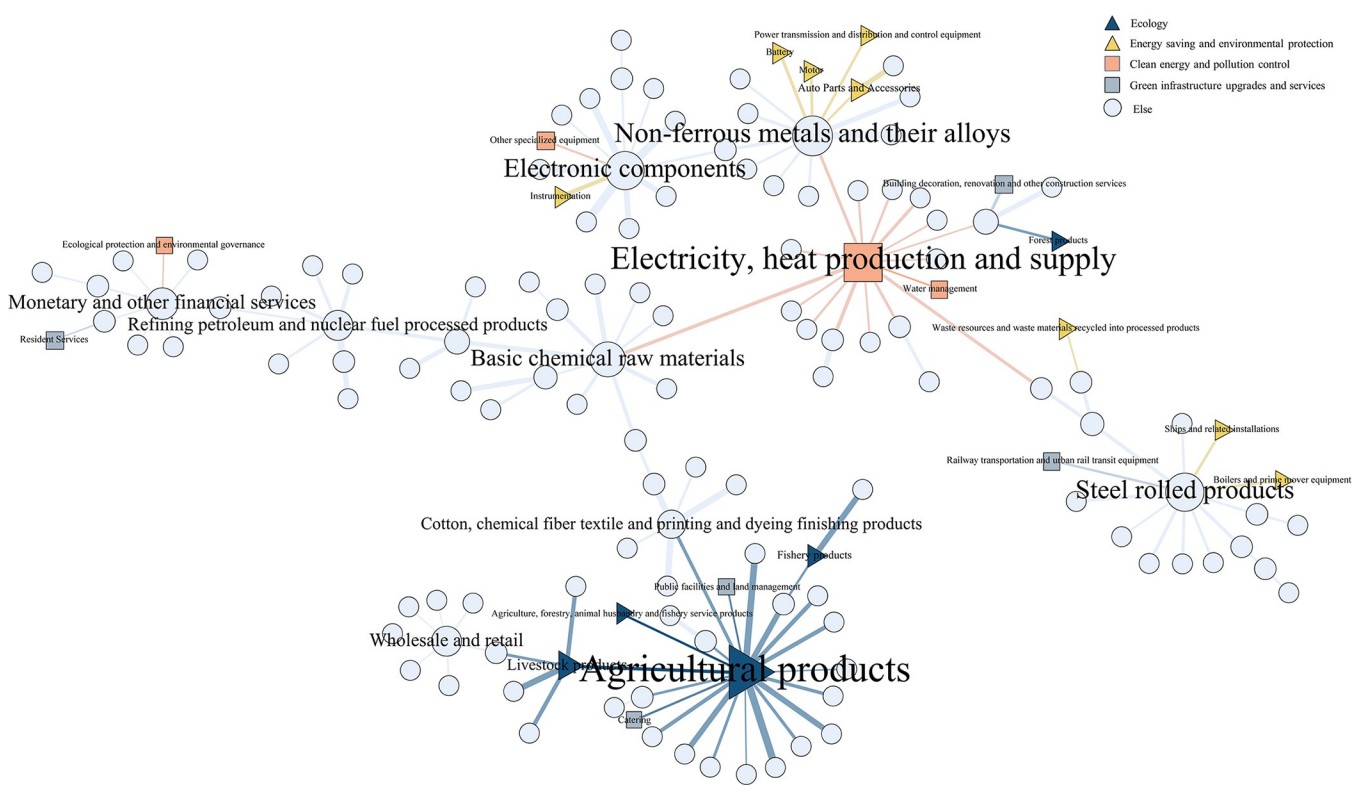

**Fig 4. Input-output table largest spanning tree in 2012.**

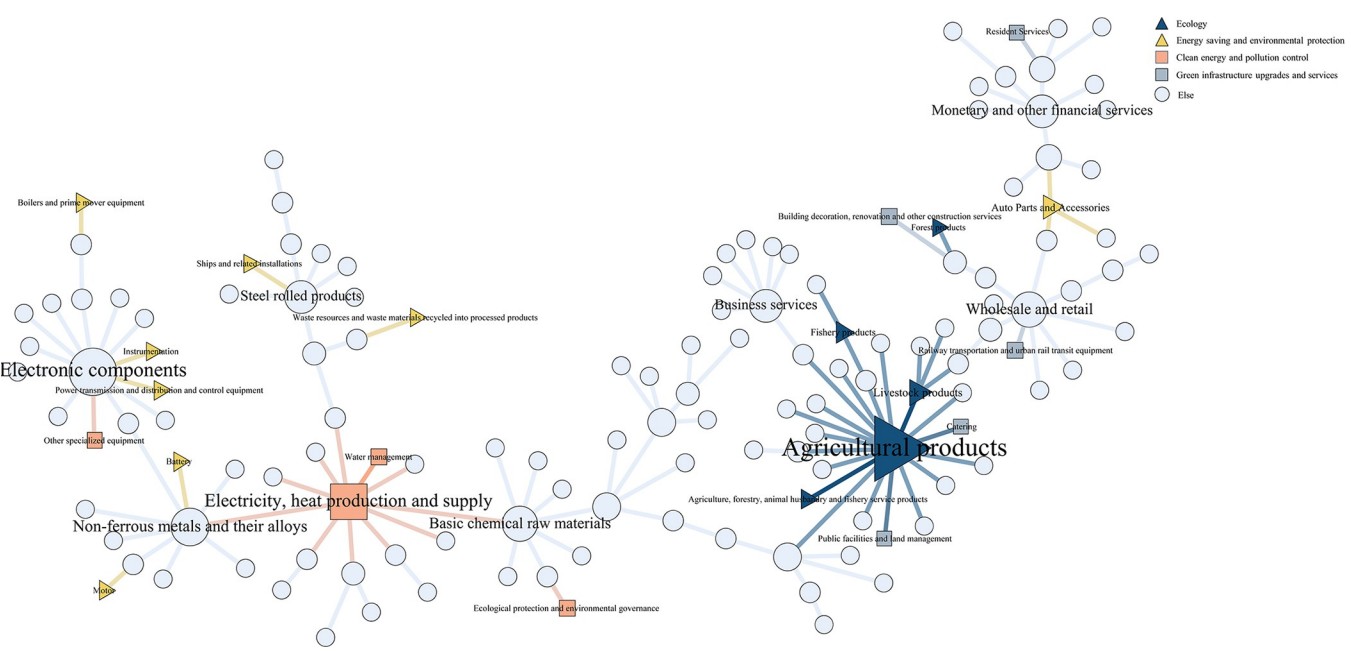

**Fig 5. Input-output table largest spanning tree in 2017.**

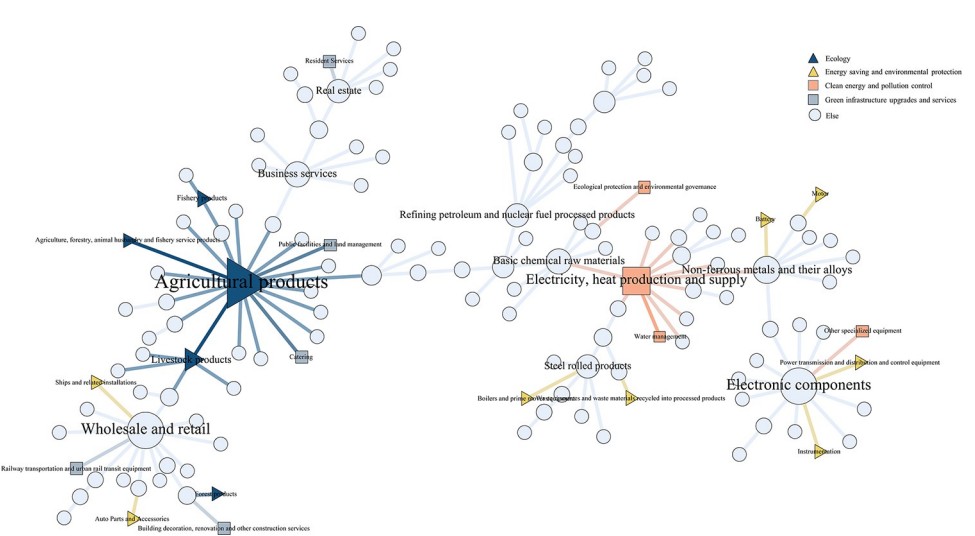

**Fig 6. Input-output table largest spanning tree in 2018.**

**Fig 7. Input-output table largest spanning tree in 2020.**

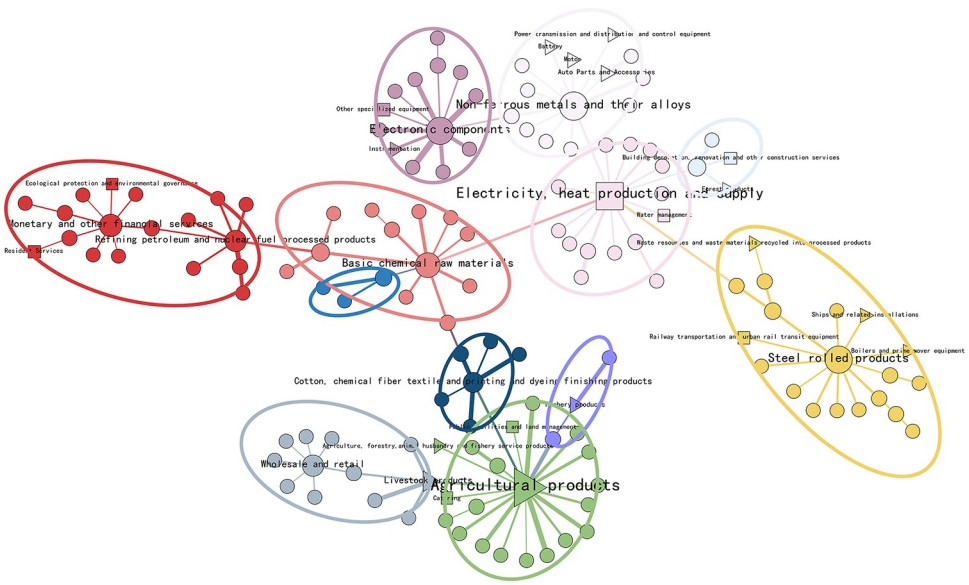

**Fig 8. Industry network community division map in 2012.**

obtained by adopting the community division algorithm. The following is an example of the industry network in 2012 and 2020, in which the green industries are labeled with triangles and squares, respectively. And the LCC algorithm is used to mine the network groups, respectively, to get 12 and 10 industrial communities, which are identified with circles in the network. The Modularity of both community networks is greater than 0.3, which indicates the rationality and reliability of the community division [56]. This is shown in Figs 8 and 9.

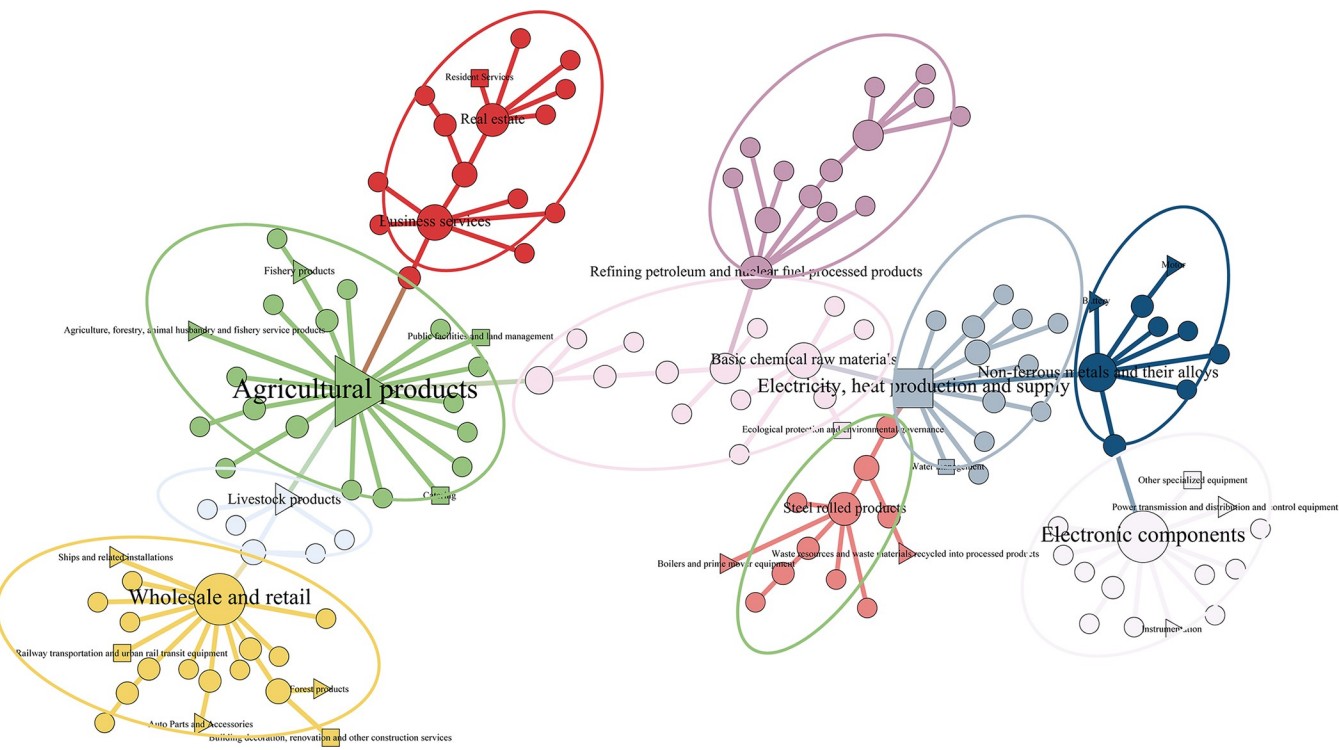

**Fig 9. Industry network community division map in 2020.**

The community analysis results show significant differences between the input-output networks of 2012 and 2020. In 2012, there was a significant variation in the number of industries within each community, while in 2020, the number of industries within each industrial community was relatively evenly distributed. Specifically, in 2012, the industrial network generated 12 industrial communities, and the number of industries within the smallest four communities did not exceed 6. In contrast, in 2020, the industrial network had 10 industrial communities, fewer than in 2012, and the two smallest communities also had more than 6 industries within them. It can be seen that the industrial clustering model in 2012 needs complexity. The industries may be relatively spatially detached, lacking effective complementary and synergistic relationships, and have yet to coalesce into a mature industrial chain. In contrast, the industrial clustering pattern in 2020 is more mature, with larger community sizes and tighter inter-industry relationships.

From the structure of the community, the industrial community in 2012 showed a tendency to cluster by types of A1-agricultural products, steel rolling products, non-ferrous metals and their alloys, basic chemical raw materials, electronic components, monetary and other financial services, and C4-electricity, heat production and supply. The clustering is more based on agriculture and manufacturing. Among them, A1-agricultural products hold a central position and are closely linked to many other industries, playing a vital hub role in connecting various industries within the industrial communities. In 2020, the industrial communities are clustered according to wholesale and retail, business services, electronic components, A1-agricultural products, and C4-electricity, heat production and supply. Compared to 2012, there is a gradual clustering trend around the tertiary service industry in the 2020 communities, and the momentum is strong. With the growth of China's economy and the upgrading of industrial structure, the clustering role played by the tertiary industry has become increasingly prominent [57].It is worth noting that the C4- electricity, heat production and supply is gradually playing a central role in the backbone network, becoming a hub node linking communities in various industries [55]. These industries are surrounded by such a large number of industries, reflecting that these industries have a strong attraction to other industries, and the resources of other industries can easily flow into these industries, thus forming industrial clusters. Community clustering is carried out based on the total consumption matrix in the input-output model analysis. It can demonstrate the flow of resources and interdependence among different industrial sectors, further helping identify key economic industries. Therefore, the central and core industries of the Chinese economy can be identified as A1-agricultural products, and C4-electricity, heat production and supply in the green industry.

In the industry categorization in the IOT, there are a total of 22 industry classifications attributed to the green economy, encompassing the four major modules of the green and energy-saving economy, namely, energy saving and environmental protection, clean energy and pollution control, green infrastructure upgrades and services, and ecology. In the mentioned graph, the green industry nodes are represented by triangles and squares. The green industry in 2012 has not yet formed a large-scale group in the network, and more green industries are scattered in each industrial community, with weak interconnections between them. In contrast, by 2020, the five industries of the industrial classification attributed to the green economy have formed an eco-friendly group centered on agricultural products on the left side of the network. In addition, the rest of the green industries have also formed groups of a specific size in the network. However, most green industries are located on the periphery of industrial communities and have yet to develop their competitive advantages. Moreover, as the status of the service industry continues to rise, it is also shaking the core position of green industries. Even though the green industry, as the center of China's economy, still has immense growth potential, it needs to be further promoted and developed.

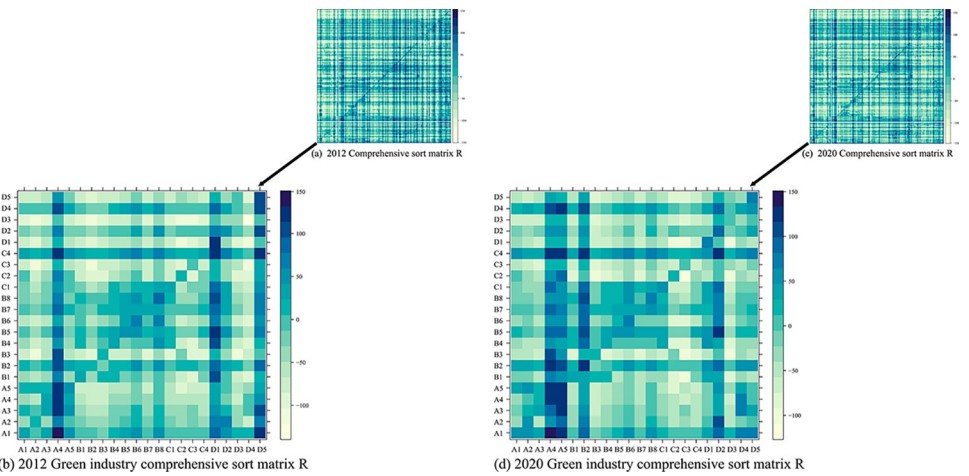

**Fig 10. Industry comprehensive ranking matrix diagram.**

**4.2.3 Selection of dominant industries.** The MST illustrates the community structure and the role of green industries in the industrial network in 2012 and 2020. Next, it is necessary to determine which green industries play a leading role in the industrial chain and which green economic development can better drive the economy forward. According to the indicator evaluation system and matrix, A of total consumption coefficients, the information entropy, weights, and the number of dominant industries for each industry input can be calculated. The comprehensive ranking matrix A and ranking values for the green industry in 2012 and 2020 can be calculated based on the formula, as shown in Fig 10 and Table 4.

Based on the comprehensive ranking matrix of industries in Fig 10, compared with 2012, the comprehensive ranking value of the input indicators of green industries in the green industry group in 2020 has been significantly improved. It is especially true for industries such as A5- agriculture, forestry, animal husbandry and fishery service products, D3- public facilities and land management, and D1-railway transportation and urban rail transit equipment. Furthermore, there are more green industries with comprehensive ranking values higher than 0,

**Table 4. Green industry composite ranking value and ranking.**

| 2012 | | | | | | 2020 | | | | | |
|---|---|---|---|---|---|---|---|---|---|---|---|
| Name | Sum | Rank | Name | Sum | Rank | Name | Rum | Rank | Name | sum | rank |
| A1 | 50.01 | 9 | C1 | 13.09 | 51 | A1 | 59.71 | 12 | C1 | 16.81 | 53 |
| A2 | 7.93 | 58 | C2 | -26.59 | 97 | A2 | 12.39 | 63 | C2 | -32.62 | 115 |
| A3 | 16.98 | 43 | C3 | -39.99 | 115 | A3 | 24.10 | 47 | C3 | -36.49 | 117 |
| A4 | -9.67 | 80 | C4 | 63.51 | 1 | A4 | 7.75 | 71 | C4 | 70.00 | 2 |
| A5 | -11.80 | 94 | D1 | -45.28 | 124 | A5 | 11.55 | 66 | D1 | -27.97 | 105 |
| B1 | -17.93 | 86 | D2 | 21.63 | 49 | B1 | -13.52 | 89 | D2 | -0.32 | 77 |
| B2 | 39.26 | 22 | D3 | -57.37 | 139 | B2 | 55.53 | 16 | D3 | -28.38 | 109 |
| B3 | -48.04 | 136 | D4 | 35.58 | 24 | B3 | -38.20 | 120 | D4 | 47.35 | 23 |
| B4 | -12.05 | 70 | D5 | -32.56 | 103 | B4 | -4.12 | 83 | D5 | -24.80 | 101 |
| B5 | 30.20 | 27 | | | | B5 | 45.17 | 25 | | | |
| B6 | -9.33 | 63 | | | | B6 | 1.15 | 76 | | | |
| B7 | 23.91 | 37 | | | | B7 | 31.58 | 36 | | | |
| B8 | 25.28 | 34 | | | | B8 | 36.20 | 30 | | - | |

indicating that their input indicators are among the dominant industries. It also implies the dominant position of more green industries within the green industry cluster, contributing to an improved internal circulation within the green industry.

According to the comprehensive industry rankings in 2012 and 2020 shown in Table 4, the dominant force of green-related industries in the economy has changed relatively little, and the status of green industries has generally shown an upward trend. Similar to other scholars' research, C4- electricity, heat production and supply and A1-agricultural products ranked high in terms of industrial influence, especially C4-electricity, heat production and supply ranked in the top three [55]. It indicates that these industries possess a dominant solid power in the economic process, which aligns with what is depicted by the MST method. However, it is worth noting that the majority of green industries have relatively weak comprehensive influence, suggesting that while green industries need to solidify their position, they also urgently need to overcome their bottlenecks and enhance the dominant power of the industry.

Overall, in the four business types of the green industry, the sum ranking of the ecology is relatively high. In contrast, the ranking of the green infrastructure upgrades and services is relatively low. On the one hand, China is a traditional agricultural country that is inseparable, and the rapid progress of national modernization at the expense of ecology in the early stage has led to the relatively weak development of China's green economy. On the other hand, China has achieved remarkable results in ecological and environmental governance, but due to the differences in high-quality development between regions, underdeveloped regions and some regions that have fallen into the dilemma of development and transformation are lagging in the adjustment of energy structure and the application of energy-saving and emission reduction technologies [1]. Efforts should focus on the ecology, energy saving and environmental protection, and clean energy and pollution control to reverse this decline. Particularly in industries with dominant solid capabilities like C4-electricity, heat production and supply, and A1-agricultural products, they should fully leverage their leadership role to drive the growth of the green industry and even the entire national economy.

## 5.Conclusion and recommendations

### 5.1 Conclusion

With the increasing emphasis on green and sustainable development worldwide, green measures such as sustainability, carbon neutrality, and peak carbon emissions are gradually being implemented. Issues related to economic development, ecological environment protection, and governance have become fundamental strategies and essential driving forces for stimulating the transformation and development of the world economy. Therefore, this study is based on four sets of input-output data from China from 2012 to 2020, combined with industrial networks and input-output methods, to construct a dynamic industrial network model to analyze the development status of China's green industry. We analyze the overall structure of industry through the topological properties and network density of the industrial network. Then, we use the MST method and the LCC algorithm to analyze the associations and cluster effects of the green economy and its industries in the industrial network. The WT model was used to evaluate the position and role of the green economy and its industries in China. The results show that:

1.The industrial structure exhibits relative stability, from the perspective of the backbone structure of the industrial complex network in China's current stage of economic development. In the early stages, the core industrial communities were primarily concentrated in A1-agricultural products, C4-electricity, heat production and supply in the agricultural and manufacturing sectors. However, starting in 2012, while agriculture and manufacturing

remained dominant, the service industry, represented by business services, witnessed remarkable growth and rapidly formed new industrial communities. It has contributed to the continued role of the green industry as a core pillar in the Chinese economy.

2.Although green industrial communities have been established, they have yet to become closely connected economic clusters and mostly remain at the periphery of the industrial network from the perspective of industrial clusters. Furthermore, China's industrial clustering has gradually moved away from the extensive clustering approach of the past. The clustering patterns have matured, with larger scales within the communities and tighter industrial linkages.

3. Green industries related to the ecology, energy saving and environmental protection, and clean energy and pollution control, hold significant importance in the sustainable development strategy from the perspective of the dominant position of the industries. They also play a crucial role in driving the economy. Among them, C4-electricity, heat production and supply, and A1-agricultural products, have the industrial solid radiative capacity, exerting significant driving effects on the entire industrial network. They directly impact the development of related industries and generate spillover effects on other industries. However, many green industries still have relatively low radiative capacity.

## 5.2 Recommendations

Based on the analysis and summary of this study, the following policy recommendations are proposed.

1. With the increasingly prominent role of the green economy in China's economic system, the status of China's green industry in the industrial structure is increasing. Therefore, it remains essential to focus on the long-term development of green industries. The green industry can increase its competitiveness and sustainable development capacity by increasing investment in green technology research and development and promoting the innovation and transformation of new energy and clean production technologies. At the same time, the government needs to formulate and improve laws and regulations to support green industries, ensure the implementation of environmental protection and energy conservation policies, and ensure their healthy development. In addition, it is also necessary to encourage enterprises and people to strengthen their awareness of environmental protection, and implement sustainable development strategies and green environmental protection concepts.

2. With the development of modern agriculture, industry, and service industry, more and higher requirements are gradually being put forward for the adjustment and optimization of green industries. China is in the critical stage of developing the "321" industry, and the characteristics of multidisciplinary interdisciplinary and field integration are becoming increasingly prominent. Green industries can improve their competitive advantages and efficiency and accelerate their transformation by introducing advanced technologies and management models. It is crucial to integrate resources, technology, and markets, foster industrial synergies, and promote the integration and clustering of industries. At the same time, the government is guiding green industries to accelerate the establishment of innovation ecosystems and industrial clusters. It is also necessary to promote cooperation and innovation between different industries, realize the sharing and optimal allocation of resources, and enhance the relevance of industrial communities.

3. The comprehensive index of the green industry indicates that the production and supply of agriculture, electricity, and heat significantly impact the overall economic network more than other industries. These areas are tangible priorities for China's green transition and economic development in the short term. Different policies and measures are needed for industries at different stages of development and trends. It is necessary for the leading industries to

consolidate and coordinate the relationship and structure between the primary industries and the upstream and downstream related industries so that they can fully play their leading role. For the fast-growing service industry, while strengthening the mechanism construction of the industry itself, it is also necessary to pay attention to the integration of the digital economy industry and the green service industry and even the related development of other industries. At the same time, for green industries with great potential and growth space for future development, it is necessary to focus on development and promotion for a long time. Such policies should be given more preferential policies, guide them to accelerate the support of technological innovation and transfer, and introduce advanced green technology and management experience to promote the linkage effect of green industries to other industries.

### 5.3 Outlook

Although this study effectively identifies the status and role of green industries and green economy in Chinese industries based on the input-output method, there are still some limitations. First, this paper analyzes the dynamic changes of green industries, but due to the data limitation that the IOT is updated every five years, the data years collected are too few to fully show the evolution process of the industrial structure. The volume of indicator data could be expanded in future studies. In addition, this study used a variety of comprehensive methods such as MST method, LCC algorithm and threshold screening to effectively identify industrial chains and key green industries, but did not further analyze the factors influencing such phenomena, which is also the direction of subsequent research. Third, we can moderately increase the research content of the spatial pattern changes and spatiotemporal comparative analysis of the development of China's green economy by conducting an overall analysis of China's green industry.

## Supporting information

**S1 File. Sample data.**
(ZIP)

## Acknowledgments

The authors thank the reviewers and the editor, whose suggestions significantly improved the manuscript.

## Author Contributions

**Conceptualization:** Yigang Guo, Jingliang Huai.

**Funding acquisition:** Shaoling Ding.

**Methodology:** Yigang Guo, Jingliang Huai.

**Project administration:** Shaoling Ding.

**Software:** Yigang Guo, Jingliang Huai, Jiayao Pan.

**Visualization:** Yigang Guo.

**Writing – original draft:** Yigang Guo, Yan Meng.

**Writing – review & editing:** Yigang Guo, Shaoling Ding.

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
