## [Decision Letter · Decision Letter 0]

3 Nov 2023

PONE-D-23-30302Multi-Model Methods for Structural Analysis of China's Green Economy Network Based on Input-Output MethodPLOS ONE

Dear Dr. Ding,

Thank you for submitting your manuscript to PLOS ONE. After careful consideration, we feel that it has merit but does not fully meet PLOS ONE’s publication criteria as it currently stands. Therefore, we invite you to submit a revised version of the manuscript that addresses the points raised during the review process.

We look forward to receiving your revised manuscript.

Kind regards,

Syed Abdul Rehman Khan, PhD

Academic Editor

PLOS ONE

Journal Requirements:

The authors acknowledge that this work was sponsored by the Guangxi Colleges and Universities Key Laboratory of Applied Statistics, and College of Science, Guilin University of Technology.

This research is funded by the National Natural Science Foundation of China，grant number 72263004；and the Natural Science Foundation of Guangxi Province，grant number 2022JJB180025. The funders had no role in study design, data collection and analysis, decision to publish, or preparation of the manuscript.

Reviewers' comments:

Reviewer's Responses to Questions

**Comments to the Author**

1. Is the manuscript technically sound, and do the data support the conclusions?

Reviewer #1: Yes

Reviewer #2: Yes

2. Has the statistical analysis been performed appropriately and rigorously? 

Reviewer #1: Yes

Reviewer #2: Yes

3. Have the authors made all data underlying the findings in their manuscript fully available?

Reviewer #1: Yes

Reviewer #2: Yes

4. Is the manuscript presented in an intelligible fashion and written in standard English?

Reviewer #1: Yes

Reviewer #2: Yes

5. Review Comments to the Author

Reviewer #1: Please present some specific figures or values of the green industries in China for the analyzed period into Introduction. Correlate your results with previou studies on this topic in China or other countries. Add limitations and directions for further research at the end of Conclusions.

Reviewer #2: General review of the manuscript

In general, this manuscript is focused on the research of green economy by using the traditional input-output method, which has good and insightful topic selection, significance emphasizing, text writing, method using, expansion studies, and conclusion summarization. Only minor revision is needed to perfect the overall content.

Detailed comments

In Introduction, please consider adding some figures & statistical graph that can visualize this research. E.g. the green economy level in different regions in China, or the proper relevant statistical data charts.

It would be better to add a research roadmap of the whole paper in the introduction.

In conclusion, please briefly re-describe the research topic and methods in this study.

In conclusion, please consider adjusting the conclusions and recommendations into 3 points respectively, and adding research deficiencies and possible future studies in your team.

Others

Please confirm that all the Figures uploaded are in high resolution.

Finally, please carefully re-check the English writing, including the sentence expressions, grammar and technical terminologies.

6. PLOS authors have the option to publish the peer review history of their article (what does this mean?). If published, this will include your full peer review and any attached files.

Reviewer #1: No

Reviewer #2: **Yes: **Penghao YE

---

## [Author Response · Author response to Decision Letter 0]

20 Dec 2023

Our changes to the article and responses to editors and reviewers are detailed in the Response to Reviewers.Please find our responses in the Response to Reviewers.

---

## [Decision Letter · Decision Letter 1]

1 Aug 2024

PONE-D-23-30302R1Multi-Model Methods for Structural Analysis of China's Green Economy Network Based on Input-Output MethodPLOS ONE

Dear Dr. Ding,

Thank you for submitting your manuscript to PLOS ONE. After careful consideration, we feel that it has merit but does not fully meet PLOS ONE’s publication criteria as it currently stands. Therefore, we invite you to submit a revised version of the manuscript that addresses the points raised during the review process.

We look forward to receiving your revised manuscript.

Kind regards,

Jiachao Peng

Academic Editor

PLOS ONE

Journal Requirements:

Reviewers' comments:

Reviewer's Responses to Questions

**Comments to the Author**

1. If the authors have adequately addressed your comments raised in a previous round of review and you feel that this manuscript is now acceptable for publication, you may indicate that here to bypass the “Comments to the Author” section, enter your conflict of interest statement in the “Confidential to Editor” section, and submit your "Accept" recommendation.

Reviewer #1: All comments have been addressed

Reviewer #2: All comments have been addressed

Reviewer #3: All comments have been addressed

Reviewer #4: (No Response)

2. Is the manuscript technically sound, and do the data support the conclusions?

Reviewer #1: Yes

Reviewer #2: Yes

Reviewer #3: Yes

Reviewer #4: Yes

3. Has the statistical analysis been performed appropriately and rigorously? 

Reviewer #1: Yes

Reviewer #2: Yes

Reviewer #3: Yes

Reviewer #4: Yes

4. Have the authors made all data underlying the findings in their manuscript fully available?

Reviewer #1: Yes

Reviewer #2: Yes

Reviewer #3: Yes

Reviewer #4: Yes

5. Is the manuscript presented in an intelligible fashion and written in standard English?

Reviewer #1: Yes

Reviewer #2: Yes

Reviewer #3: No

Reviewer #4: Yes

6. Review Comments to the Author

Reviewer #1: No further comments. The comments were properly adressed and I believe that he paper can be published in the current form.

Reviewer #2: After the review & modification works, the review comments have been seriously considered by the author(s). The manuscript R1 now shows sound topic expression, academic writing, logical essay structure, and adequate quantitative methods. Most of the deficiencies have been solved, thereafter, I personally do not have any more comments for the manuscript R1.

Summing up the above, my personal recommendation is ACCEPT. If accepted and in the proofreading stage, please re-check carefully on the manuscript writing, grammar, expressions, and possible omitted interpretations of the research topics, method descriptions and results discussion.

Reviewer #3: The authors carefully revised the questions raised by the reviewers. The revised manuscript has achieved the expected effect and met the basic requirements of the journal.

Reviewer #4: This study uses industry networks, the maximum spanning tree (MST) method, the Leiden community clustering (LCC) algorithm, and the Weaver-Thomas (WT) model to analyze the contribution and status of China's green economy in economic development. The data were obtained from the input-output tables published by the National Bureau of Statistics of China in 2012, 2017, 2018 and 2020. The industry network was constructed by internal flow matrix and total consumption coefficient matrix, and topological characterization, MST method to extract backbone network structure, and LCC algorithm for network community analysis were applied. Finally, the WT model is used to determine the importance of industry nodes and analyze their position in China's national economic system. However, the manuscript still has some issues to be improved.

1. Please further explain the theoretical and practical significance of this study.

2. What is the basis for the article's selection of industry network, maximum spanning tree (MST) method, Leiden community clustering (LCC) algorithm and Weaver-Thomas (WT) model? What are the applicability and limitations of these methods?

3. The selection of parameters and variables involved in the calculations lacks sufficient theoretical support and empirical basis.

4. The resolution of some graphs is low, especially in Figures 8 and 9, where the images appear blurred. Please adjust the resolution of the graphs according to the submission guidelines of this journal to ensure the clarity of the screen display. In addition, it is recommended that the text section be aligned at both ends.

5. The countermeasures proposed in the article are not targeted enough. Please propose practical and instructive countermeasures based on the findings of the study.

7. PLOS authors have the option to publish the peer review history of their article (what does this mean?). If published, this will include your full peer review and any attached files.

Reviewer #1: No

Reviewer #2: **Yes: **Penghao YE

Reviewer #3: No

Reviewer #4: No

---

## [Author Response · Author response to Decision Letter 1]

6 Aug 2024

All comments from specific reviewers can be found in the Response to Reviewers file.

---

## [Decision Letter · Decision Letter 2]

21 Aug 2024

Multi-Model Methods for Structural Analysis of China's Green Economy Network Based on Input-Output Method

PONE-D-23-30302R2

Dear Dr. Ding,

We’re pleased to inform you that your manuscript has been judged scientifically suitable for publication and will be formally accepted for publication once it meets all outstanding technical requirements.

Kind regards,

Jiachao Peng

Academic Editor

PLOS ONE

Additional Editor Comments (optional):

Reviewers' comments:

Reviewer's Responses to Questions

**Comments to the Author**

1. If the authors have adequately addressed your comments raised in a previous round of review and you feel that this manuscript is now acceptable for publication, you may indicate that here to bypass the “Comments to the Author” section, enter your conflict of interest statement in the “Confidential to Editor” section, and submit your "Accept" recommendation.

Reviewer #2: All comments have been addressed

Reviewer #3: All comments have been addressed

Reviewer #4: All comments have been addressed

2. Is the manuscript technically sound, and do the data support the conclusions?

Reviewer #2: Yes

Reviewer #3: Yes

Reviewer #4: Yes

3. Has the statistical analysis been performed appropriately and rigorously? 

Reviewer #2: Yes

Reviewer #3: Yes

Reviewer #4: Yes

4. Have the authors made all data underlying the findings in their manuscript fully available?

Reviewer #2: Yes

Reviewer #3: (No Response)

Reviewer #4: Yes

5. Is the manuscript presented in an intelligible fashion and written in standard English?

Reviewer #2: Yes

Reviewer #3: No

Reviewer #4: Yes

6. Review Comments to the Author

Reviewer #2: The manuscript R2 is now well-written after 2 rounds of amendments, where all the literal content and model analysis have been properly optimized. Please carefully check again all the details at the proofreading stage.

Reviewer #3: Thanks to the authors for their careful revision work. The revision is satisfactory.The manuscript meets the academic standards of the journal.

Reviewer #4: After the review & modification works, the review comments have been seriously considered by the author(s). My personal recommendation is ACCEPT.

7. PLOS authors have the option to publish the peer review history of their article (what does this mean?). If published, this will include your full peer review and any attached files.

Reviewer #2: **Yes: **Penghao YE

Reviewer #3: No

Reviewer #4: No

---

## [Editor Report · Acceptance letter]

26 Aug 2024

PONE-D-23-30302R2 

PLOS ONE

Dear Dr. Ding, 

I'm pleased to inform you that your manuscript has been deemed suitable for publication in PLOS ONE. Congratulations! Your manuscript is now being handed over to our production team.

Kind regards, 

on behalf of

Dr. Jiachao Peng 

Academic Editor

PLOS ONE